# Exosomes in Cardiovascular Disease: From Mechanism to Therapeutic Target

**DOI:** 10.3390/metabo13040479

**Published:** 2023-03-27

**Authors:** Allison B. Reiss, Saba Ahmed, Maryann Johnson, Usman Saeedullah, Joshua De Leon

**Affiliations:** Department of Medicine and Biomedical Research Institute, NYU Long Island School of Medicine, Mineola, NY 11501, USA

**Keywords:** atherosclerosis, cholesterol, cardiovascular disease, microRNA, exosome, oligonucleotide

## Abstract

Cardiovascular disease (CVD) is the leading cause of morbidity and mortality globally. In recent decades, clinical research has made significant advances, resulting in improved survival and recovery rates for patients with CVD. Despite this progress, there is substantial residual CVD risk and an unmet need for better treatment. The complex and multifaceted pathophysiological mechanisms underlying the development of CVD pose a challenge for researchers seeking effective therapeutic interventions. Consequently, exosomes have emerged as a new focus for CVD research because their role as intercellular communicators gives them the potential to act as noninvasive diagnostic biomarkers and therapeutic nanocarriers. In the heart and vasculature, cell types such as cardiomyocytes, endothelial cells, vascular smooth muscle, cardiac fibroblasts, inflammatory cells, and resident stem cells are involved in cardiac homeostasis via the release of exosomes. Exosomes encapsulate cell-type specific miRNAs, and this miRNA content fluctuates in response to the pathophysiological setting of the heart, indicating that the pathways affected by these differentially expressed miRNAs may be targets for new treatments. This review discusses a number of miRNAs and the evidence that supports their clinical relevance in CVD. The latest technologies in applying exosomal vesicles as cargo delivery vehicles for gene therapy, tissue regeneration, and cell repair are described.

## 1. Introduction

Cardiovascular disease (CVD) is a leading cause of morbidity and mortality worldwide [1,2]. It is associated with enormous health and economic burdens [3,4,5]. Despite current risk assessment and treatment modalities, residual CVD hazard remains unacceptably high [6,7]. Given its substantial presence in society, extensive research to deepen our understanding of the underlying mechanisms of CVD initiation and progression are imperative. It has been increasingly recognized that extracellular vesicles (EVs), specifically exosomes, have a significant role in CVD development and pathology. Over the last decade, the perception of exosomes as cell debris derivatives with no significant impact on neighboring and distant cells has changed drastically [8,9]. These minuscule cellular biological materials are particularly important in cell–cell communication, information transfer, and cell function. Exosomes are readily accessible and available from various tissues throughout the body and travel freely through the circulation. They exhibit varying effects via their contents depending on the type and health of their cell of origin. Due to their unique characteristics and dynamics, they are key factors in regulating cardiovascular function and play an important role in almost all aspects of cardiovascular pathology [10]. The various micro (mi)RNAs carried by exosomes directly influence multiple aspects of CVD such as hypertrophy, angiogenesis, apoptosis, fibrosis, injury, and repair. Given their diverse role in distinct aspects of CVD, these microvesicles can be leveraged in the diagnosis and treatment of the disease [11]. Their potential as therapeutic agents and as nanocarriers and protectors of biomolecules has raised interest among researchers and healthcare professionals. This review probes many aspects of this critical link between exosomes and CVD [12].

## 2. Overview of Exosomes

Intercellular communication among cells is vital in all living organisms for normal functioning [13,14]. Cell-to-cell communication occurs through direct contact in neighboring cells or via secretion of soluble factors, such as cytokines, hormones, and chemokines, which can extend interaction over distances [13,15]. Recent attention has focused on a means of intercellular communication via extracellular vesicles (EVs), cell-derived membrane-enclosed microvesicles of varying sizes secreted by most cell types [16,17]. EVs show promise in a number of medical applications and may be useful as non-invasive diagnostic biomarkers and therapeutic nanocarriers for the treatment of various diseases, including neurodegeneration, cardiovascular dysfunction, and cancer [18,19,20,21]. EVs can be categorized into four major classes, microvesicles, apoptotic bodies, virus-like particles, and exosomes, according to their size, subcellular origin, and content [22,23]. While the first three classes of EVs are formed by the outward budding of the plasma membrane, exosomes originate as intraluminal vesicles contained within multi-vesicular bodies (MVBs) (Figure 1) [24,25]. Successive extracellular secretion of these intraluminal vesicles by fusion with the plasma membrane releases exosomes with diameters of ~40 to 160 nanometers into the extracellular matrix [12,20,26].

Exosomes released into surrounding bio-fluids mediate cell–cell communication between adjoining and distant cells by shuttling cell-specific cargo composed of bioactive molecules such as lipids, carbohydrates, metabolites, surface and cytoplasmic proteins, and nucleic acids [27,28]. Exosome cargo is subject to selective sorting, is unique to the cell type from which they are derived and reflects the functions and current state of the cell of origin [29,30]. Exosomes are also enriched in specific proteins such as tetraspanins that distinguish them from their cell of origin and can be used to track and identify them [31]. Their lipid bilayer structure is well-suited to carrying and protecting their contents over time and distance so that they can be delivered intact from donor to receptor cell and from one part of the organism to the other.

## 3. The Link between Cardiovascular Disease and Exosomes

### 3.1. Exosome Cargo and the Cardiovascular System

Exosomes are key factors in regulating CVD progression due to their role in the transport and exchange of signaling molecules [32,33,34,35]. A study by Peng et al. found that extracellular vesicles from atherosclerotic plaque with characteristics of exosomes may be able to spread atherosclerosis distally [36]. Extracellular vesicles were isolated from carotid atherosclerotic lesions of rats with knockout of the LDL receptor gene on a high-fat diet and injected into LDL receptor knockout mice on a normal diet, conditions under which they do not normally develop atherosclerosis. The extracellular vesicles from atherosclerotic tissue were taken up by endothelium in the carotids and promoted endothelial inflammation.

In the heart and vasculature, cell types that interact to maintain homeostasis and are known to release exosomes include cardiomyocytes, endothelial and vascular smooth muscle cells, cardiac fibroblasts, inflammatory cells, and resident stem cells (Figure 2) [35,37,38,39]. Under both physiological and pathological conditions, exosomes are an integral mode of communication amongst these cell types (Table 1). Their rate of secretion and specific miRNA cargo can change in response to pathological states or drug treatments [40,41,42,43]. For example, in a mouse model, analysis of the miRNA content of exosomes derived from EPC found that the top ten miRNAs in abundance were all associated with atherosclerosis [44]. Administration of EPC-derived exosomes to atherosclerosis-prone hyperglycemic mice lowered oxidative stress and inflammation and reduced plaque area, indicating that EPC exosomes may improve endothelial function in diabetic atherosclerosis.

Further supporting exosome influence on atherosclerosis of specific exosome cargo from distinct cell types, cardiomyocyte exosomes are enriched for heat shock proteins (HSP) such as HSP-20, 60, and 70) [45,46,47]. HSPs are molecular chaperones that aid in protein folding and take part in apoptotic processes, cell proliferation, and inflammatory responses [48]. A higher expression of HSP-20 has cardioprotective effects, likely via the inhibition of TNF-α and IL-1β [46,49]. HSP-60, localized within mitochondria, supports mitochondrial function and protein homeostasis under stress conditions, but when released extracellularly, is apoptotic, inflammatory, and is considered destructive and atherogenic [50,51]. HSP-70 is found on the exosome surface and, by binding to toll-like receptor-4 on the cardiomyocyte, it can activate the MAPK/ERK1/2 signaling pathway leading to pro-survival and cytoprotective effects [52,53].

Exosomal HSP-27 release can also affect macrophage cholesterol homeostasis [54,55]. HSP-27 is considered atheroprotective, and low levels of this protein in plasma are associated with CVD [56]. Shi et al. combined HSP-27 with an anti-HSP-27 antibody to form immune complexes and found that exposure of cholesterol-loaded THP-1 human macrophages to these immune complexes augmented release of exosomes and these exosomes were enriched in cholesterol, thus facilitating macrophage cholesterol efflux [57]. They propose the possibility of packaging HSP-27 immune complexes in exosomes as a means of delivering immune therapy against atherosclerosis to macrophages.

Exosomes can also carry lipids, including free fatty acids, which Barcia et al. showed can be delivered to the heart in vivo and to cardiac cell types in vitro [58,59]. This group isolated serum exosomes from healthy adults under fasting conditions and after consumption of a high-calorie meal (post-prandial) and found that each could take up a free fatty acid analogue, but the post-prandial exosomes had higher levels of the scavenger receptor CD36 content and higher lipid content. Blocking CD36 reduced free fatty acid analogue uptake, indicating an active role for this receptor in exosomes. In cell culture experiments, cardiac endothelial cells and cardiomyocytes were able to take up the free fatty acid analogue from serum exosomes and the mouse heart was able to take up the analogue after tail vein injection of the exosomes, leading the authors to posit a role for exosomes in delivering this type of lipid fuel to the heart.

### 3.2. Exosomes in Atherosclerosis: Endothelial Dysfunction, Macrophage Recruitment, and Vascular Smooth Muscle Behavior

Endothelial dysfunction is an early step in the development of atherosclerosis. Activated endothelial cells upregulate expression of adhesion molecules ICAM-1, VCAM-1, P-selectin, and E-selectin, which act to recruit monocytes to the subendothelial layer of the artery. Important risk factors for atherosclerosis that cause activation of endothelium are an elevation in serum levels of oxidized LDL (ox-LDL) and an inflammatory environment of elevated cytokines and homocysteine [60,61,62,63]. Ox-LDL and homocysteine have been shown to induce the release of HSP-70-containing exosomes from aortic endothelial cells that can elicit immune inflammatory responses and selectively activate monocytes [64,65]. Monocyte activation leads to monocyte-endothelial cell adhesion and monocyte infiltration into the subendothelial space. Activated monocyte-derived macrophages in the arterial intima orchestrate multiple inflammatory atherosclerosis-promoting processes in plaque formation [66]. When activated, monocytes themselves can perpetuate their own adhesion to endothelium by producing exosomes that enter endothelial cells and activate NF-κB, thus causing endothelial production of adhesion molecules [67]. Exosomes isolated from murine bone marrow-derived macrophages (BMDM) and infused into atherosclerosis-prone ApoE-deficient Western diet-fed mice exhibit an ability to reduce atherosclerotic lesion necrosis and stabilize atheroma [68]. Cheng et al. showed, in a human cell culture model, that human umbilical vein endothelial cells cultured under inflammatory conditions were protected from apoptosis by exosomes from M2 polarized THP-1 macrophages and the effect was mediated by miR-221-3p [69].

VSMC proliferation, migration, and cytokine secretion contribute to atherogenesis [70]. VSMC behavior can be influenced by exosomes of macrophage origin. As far back as 2016, Niu et al. showed that extracellular vesicles derived from J774a.1 macrophage foam cells could enhance migration and adhesion of human aortic VSMC [71]. Ren et al. cultured VSMC in media with exosomes from ox-LDL-stimulated macrophages and found that these exosomes improved viability and invasive properties of the VSMC while suppressing apoptosis [72]. Of the overexpressed miRNA in the ox-LDL-stimulated macrophage exosomes, miR-186-5p was found to be responsible for these pro-atherogenic effects via inactivation of SHIP2, thus enhancing PI3K/AKT/mTOR signaling.

The influence of VSMC-derived exosomes on atherosclerosis has been studied. Exosomes isolated from cultured primary human VSMC induced to undergo premature senescence and their properties were then analyzed [73]. The senescent VSMC produced more exosomes than the control VSMC. When T cells and monocytes were exposed to exosomes of senescent VSMC origin, they produced more pro-inflammatory cytokines than cells exposed to control VSMC exosomes.

### 3.3. Exosomes in Apoptosis

A critical event in CVD progression, including myocardial infarction and heart failure, is the apoptosis and autophagy of cardiomyocytes [74]. Myocardial ischemia followed by therapeutic restoration of blood flow (ischemia-reperfusion) leads to cardiomyocyte damage and apoptosis in both the hypoxia stage and the rapid recovery stage [75,76]. Cardiac fibroblasts release exosomes that rescue cardiomyocytes from ischemia-reperfusion injury by protecting against apoptosis and pyroptosis (a form of programmed necrosis) [52,77,78]. Luo et al. showed that injection of rat cardiac fibroblast exosomes into the infarct border of the rat heart during hypoxia–reoxygenation (left anterior descending artery ligation and reperfusion) reduced infarct size. Comparison of expression levels of exosomal miRNAs showed miR-423-3p to be enriched in exosomes that limited infarct size and knockdown of this miRNA in cell culture companion studies confirmed its importance in maintaining cell viability and reducing apoptosis [79].

In studies using human exosomes, Qiao et al. isolated exosomes from conditioned media of cardiac cells from healthy volunteers and heart failure patients and performed intramyocardial injection of these exosomes into a mouse model of acute myocardial infarction induced by coronary vessel ligation. The study showed that the healthy volunteer exosomes had favorable effect on healing, apoptosis, and tissue preservation while exosomes of heart failure patients inhibited cardiomyocyte proliferation, inhibited angiogenesis, and worsened healing [80]. As in the case of exosomes from heart failure patients, exosomes from EPC of mice deficient in the anti-inflammatory and atheroprotective cytokine IL-10 also lose their protective effect in reducing infarct damage and, rather than inhibit apoptosis, these exosomes enhanced apoptosis [81,82].

### 3.4. Exosomes in Hypertrophy

Pathologic cardiac hypertrophy occurs when the wall of the ventricle thickens, impairing systolic function [83]. It is often a result of chronic hypertension and can lead to ischemia and eventually to heart failure. Hallmarks of this process are enlargement of cardiomyocytes and fibrosis. Exosomes have been implicated in the regulation of cardiac hypertrophy via effects on the renin-angiotensin system. The mechanical stretch of cardiomyocytes has been found to release exosomes enriched with angiotensin II and its receptor content, contributing to increased vascular resistance and hypertrophy [84,85,86].

Constantin et al. found a potential benefit of exosomes from human mesenchymal stem cells in ameliorating hypertrophy in a model of human hypertrophic cardiomyocytes [87]. When hypertrophic cardiomyocytes were incubated with extracellular vesicles from subcutaneous adipose tissue stem cells or from bone marrow mesenchymal stem cells, cardiac markers for cardiomyocyte hypertrophy and inflammatory cytokine such as IL-1β, IL-4, IL-6, and TNF-α decreased [88].

Mao et al. engineered exosomes derived from human cardiosphere-derived cells to target cardiomyocytes [89]. They then infused these exosomes into a mouse model of myocardial hypertrophy and showed decreased perivascular and myocardial interstitial fibrosis of the left ventricle and decreased hypertrophy in treated mice versus sham mice and attributed the differences largely to effects of miRNA-148a, an miRNA with anti-proliferative properties.

### 3.5. Exosomes in Angiogenesis

The regulation of blood vessels involved in supporting the myocardium with oxygen and nutrients relies, in part, on communication between cardiomyocytes and myocardial vascular endothelial cells via exosomes [90]. Angiogenic capacity is of great importance in cardiac repair and regeneration post-ischemic injury. A number of cell types can release exosomes that deliver specific pro-angiogenic miRNAs. For example, mesenchymal stem cell exosomes isolated from human adipose tissue have been shown to promote angiogenesis in human umbilical vein endothelial cells by carrying miR-125a [91]. Exosomes extracted from cardiomyocytes after cardiomyocyte hypoxic preconditioning can protect cardiac microvascular endothelial cells from oxidative damage and promote angiogenesis in vitro and in vivo and this effect is mediated predominantly by miR-222 and miR-143 [92,93]. Gou et al. showed that purified exosomes from primary neonatal cardiomyocytes treated with H2O2 to induce injury were enriched in miR-19a-3p and this increase in miR-19a-3p was also true for human plasma in post-myocardial infarction patients [94]. They demonstrated uptake of exosomes from mouse neonatal cardiomyocytes by endothelial cells and showed that miR-19a-3p was anti-angiogenic and that neutralizing this miRNA promoted survival and proliferation of endothelial cells. They performed mouse studies in which they induced myocardial infarction via ligation of the left anterior descending coronary artery and then silencing of miR-19a-3p and showed that silencing improved angiogenesis. The target of miR-19a-3p was found to be hypoxia-inducible factor (HIF)-1α, which is known to drive multiple genes involved in angiogenesis [95]. In a rat model of myocardial infarction induced by ligation of the left anterior descending coronary artery, exosomes derived from cardiac telocytes enhanced angiogenesis and reduced cardiac fibrosis [96]. Increased miR-126 and miR-199a in circulating exosomes in patients with coronary artery disease is associated with a reduced risk for major future adverse cardiovascular events [97]. Cultured endothelial cells exposed to these exosomes showed increased proliferation, migration, and tube-formation [98,99].

In the microenvironment induced by myocardial infarction, M1 macrophages predominate early, are inflammatory, and generate exosomes that transport miRNAs to endothelial cells that can reduce angiogenic potential and aggravate myocardial injury [100]. These proinflammatory exosomes are enriched in miR-155, which is known to inhibit pathways within endothelial cells, such as the Sirtuin 1, adenosine monophosphate-activated protein kinase (AMPK), and endothelial nitric oxide synthase which promote angiogenesis [101,102].

### 3.6. Exosomes in Cardiac Fibrosis

Cardiac fibroblasts play an integral role in normal myocardial structure and physiology [103]. They produce an extracellular matrix and pro-fibrotic factors that can be reparative following injury [104]. However, when pathologically activated, they negatively affect cardiac function as well as myocardial compliance and stiffness [105,106,107,108]. Exosomes derived from different cell types and originating under normal or pathologic conditions can affect fibroblasts and modulate cardiac fibrosis. For instance, in a mouse model of diabetes, exosomes released from the heart after exercise on a treadmill reduced fibrosis via their higher content of miR-29b and miR-455, which acted by downregulating pro-fibrotic matrix metalloproteinase (MMP)-9 [109]. In humans with heart failure, the level of miR-217 in cardiac tissue is elevated and higher miR-217 correlates with poorer left ventricular ejection fraction [110,111]. In a mouse model of heart failure induced by thoracic aortic constriction, cardiomyocyte-derived exosomes enriched in miR-217 aggravated cardiac fibrosis via targeting of phosphatase and tensin homolog (PTEN) [111]. Yuan et al. also used the thoracic aortic constriction mouse model to look at cardiac remodeling under mechanical stress and found inhibition of excessive fibrosis in mice given an miR-378 mimic. Exosomes from cultured cardiomyocytes that had been subjected to mechanical stress harbored miR-378 and when cardiac fibroblasts that had also been subjected to mechanical stress were exposed to these exosomes, they showed reduced upregulation of collagen expression and lower MMP9 protein levels [112]. Anti-fibrotic effects on the cardiac fibroblasts were blocked with depletion of miR-378 from the stretched cardiomyocyte conditioned medium. Using murine cells in culture, Tang et al. showed that mechanical stress-treated cardiomyocytes can activate cardiac fibroblasts by delivering miR-494-3p in exosomes [113]. The cardiomyocyte exosomal miR-494-3p targets PTEN in the fibroblasts. The miR-494-3p enrichment of cardiomyocyte exosomes occurs due to stress-induced upregulation of the E3 ubiquitin ligase Peli1, a known participant in inflammatory processes, and is abrogated in cardiomyocytes that are deficient in Peli1.

The pathological activation of cardiac fibroblasts is also influenced by exosomes derived from macrophages. In the diabetic microenvironment, exosomes released from macrophages exacerbate cardiac fibrosis [114].

### 3.7. Exosomes in Myocardial Infarction

Exosomes are considered potential biomarkers for the diagnosis and monitoring of myocardial ischemia and infarction [115,116]. This is of value because of the limitations in sensitivity and specificity and delay in detectable change in the standard marker, troponin T [117]. Earlier diagnosis and confirmation of myocardial infarction allows faster intervention for heart muscle preservation [118].

Silvia-Palacios conducted a study in Mexico City in which exosomes were isolated from the plasma of 26 patients admitted to intensive care with acute ST-segment elevation myocardial infarct (STEMI) and compared to those of 26 healthy donors [119]. They found that ischemia prompted increased exosome release from the myocardial infarction patients. Both during the acute myocardial infarction and after reperfusion, miR-223-3p, an miRNA that targets inflammatory molecules, was found at high levels in patient exosomes.

Chen et al. looked at differentially expressed miRNAs in plasma exosomes from patients with STEMI and non-STEMI versus healthy controls and proposed a set of 10 miRNAs that they found could discriminate between STEMI and non-STEMI, which has implications for prognosis and treatment planning [120].

### 3.8. Adipose Tissue Exosomes

Obesity is often associated with atherosclerosis and the metabolic complications of obesity may implicate adipose-tissue (AT)-derived exosomes, although their explicit role in atherogenesis remains unclear [121,122]. Adipocyte-derived exosomes have been shown to deliver adipocyte-dominant transcripts into macrophages and promote AT macrophage activation [123,124]. Xie et al. showed that exosomes released from differently located and stressed ATs have distinct pro-atherosclerotic effects [121]. Barberio et al. characterized miRNAs from human adipose tissue-derived exosomes from obese and lean adolescents and found that six miRNAs contained in adipocyte exosomes targeted cholesterol efflux genes and were significantly associated with cholesterol efflux capacity [125]. They then exposed cultured THP-1 macrophages to adipocyte-derived exosomes and showed that obese subject exosomes significantly increased macrophage ox-LDL retention compared to lean subject exosomes. The findings of this study exemplify a mechanistic link between obesity and macrophage lipid handling.

Liu et al. showed, in mice, that perivascular adipose tissue exosomes suppress macrophage foam cell transformation [126]. Treatment of a mouse macrophage cell line with the perivascular adipose tissue exosomes reduced ox-LDL uptake, likely due to decreased expression of ScR-A mRNA and promoted cholesterol efflux, likely due to enhanced expression of ABCA1 and ABCG1 mRNAs. They then conducted further study of the mechanisms involved and found that the key miRNA mediating the upregulation of macrophage ABCA1 and ABCG1 by perivascular adipose tissue exosomes was miR-382-5p [126].

Evidence is beginning to accumulate that exosomes from different cell types found within obese adipose tissue affects whole body metabolism, and, indirectly, atherosclerotic risk [127,128,129,130]. For example, macrophages that reside within adipose tissue produce exosomes that transfer to adipocytes, and, in a murine model, lean mice treated with adipose tissue macrophage exosomes from obese mice became insulin resistant [131]. Further evidence of the relationship among obesity, adipose tissue exosomes, and metabolism can be found in a human study showing that weight loss brought about after gastric bypass surgery changes the miRNA content of adipocyte-derived exosomes isolated from the peripheral blood [132]. One year after surgery, 10 miRNAs that target insulin signaling pathways were found to be altered and this correlated with improvements in insulin sensitivity with better glucose homeostasis.

**Table 1 metabolites-13-00479-t001:** Summary of cardiovascular impact of exosome cargo from specific cell and tissue types.

Cell/Tissue Source of Exosomes	Reference	CVD Impact	Key Cargo Involved
Carotid atherosclerotic tissue	[36]	Endothelial inflammation in rats	Whole exosomes
Endothelial progenitor cells (EPC)	[44]	↓ oxidative stress and inflammation and ↓ plaque area in hyperglycemic mouse model	Whole exosomes
Cardiomyocyte	[49]	Protects against apoptosis, remodeling and cardiac hypertrophy in diabetic mice	Heat shock protein (HSP)-20,
Cardiomyocyte	[50,51]	Atherogenic via induction of inflammatory mediators	HSP-60
Blood	[56,57]	Anti-inflammatory, atheroprotective effects on macrophages via IL-10 release, facilitation of cholesterol efflux	HSP-27
Blood	[59]	Delivery of free fatty acids to cardiac endothelium and myocytes for energy	CD36
Arterial endothelium	[64,65]	Monocyte activation and adhesion	HSP-70
Macrophage foam cells	[71,72]	Enhanced adhesion and suppressed apoptosis of vascular smooth muscle cells	MiR-186-5p
Cardiac fibroblasts	[77,78,79]	↓ infarct size in rat ischemia-reperfusion injury, limited apoptosis	MiR-423-3p
Human mesenchymal stem cells	[88]	↓ cardiac markers for cardiomyocyte hypertrophy and ↓ inflammatory cytokine levels	Whole exosomes
Human cardiospheres	[89]	↓ left ventricular fibrosis and hypertrophy in a mouse cardiac hypertrophy model	MiRNA-148a
Human adipose tissue mesenchymal stem cells	[91]	Promote angiogenesis	MiR-125a
Cardiomyocyte after hypoxic preconditioning	[92,93]	Promote angiogenesis and protect microvascular endothelium from oxidative damage	MiR-222 and miR-143
Primary neonatal cardiomyocytes	[94,95]	Anti-angiogenic	MiR-19a-3p
Blood	[97,98,99]	↑ proliferation, migration and tube-formation in endothelial cells	MiR-126 and miR-199a
Inflammatory M1 macrophages	[101,102]	Anti-angiogenic	MiR-155
Heart after exercise	[109]	↓ fibrosis	MiR-29b and miR-455
Cardiomyocyte	[111]	↑ fibrosis and hypertrophy	MiR-217
Cardiomyocytes after mechanical stress	[112]	↓ fibrosis	MiR-378
Cardiomyocytes after mechanical stress	[113]	↑ fibrosis	MiR-494-3p
Perivascular adipose tissue	[126]	Atheroprotective, ↑ macrophage cholesterol efflux, ABCA1, and ABCG1	MiR-382-5p
Obese mouse adipose tissue macrophages	[131]	Confer insulin resistance on lean mice	Whole exosomes

## 4. Exosomal MiRNAs of Specific Interest in CVD

The most fundamental biological cargo within exosomes are miRNAs and evidence is accumulating that they are specifically critical in cardiac cell communication [133]. The content of the specific exosomal miRNAs secreted with or without stimulus from the heart and vasculature changes based on the pathophysiological settings of heart disease. Therefore, determining the circulating levels and distinguishing the sequences of exosomal miRNAs can aid in the diagnosis, monitoring, and treatment of CVD [134]. The over or under expression of specific miRNAs has been found to promote atherogenic conditions and the miRNAs involved in these processes may therefore be considered targets for therapeutic intervention. The candidate miRNAs for atherosclerosis treatment are numerous and the list is increasing rapidly. Here we have selected for discussion the miRNAs with the greatest potential clinical relevance in CVD. Of necessity, this paper will discuss a few standout examples that hold promise (Table 2). Human clinical trials are lacking, so much of the data is from murine studies. We performed a literature review using PubMed and Google Scholar, including the search terms “(atherosclerosis, atherosclerotic cardiovascular disease OR ASCVD) AND (microRNA OR miR OR miRNA)” from 2010 to the present, written in English-language journals. We then performed this original search with the addition of the term “AND obesity” in order to find recent obesity-related articles and we performed the original search with the addition of the term “AND human” to find studies using human subjects.

Although this list is not comprehensive, it highlights the exosomal miRNAs with the most robust clinical relevance either because the data are derived from human or primate studies or because these miRNAs affect pathways that are clearly related to atherosclerosis. This is a curated list rather than a comprehensive one and incorporates obesity because the current literature relating to adipose tissue characteristics, metabolic dysfunction, and CVD is strong and growing and affects millions around the globe [135,136].

### 4.1. MiR-19b

Mouse models have shown the important mechanisms through which miRNA can influence the course of pathological conditions such as atherosclerosis. An in vivo murine study showed how elevated expression of miR-19b can lead to the progression of atherogenesis. Based on human subject studies showing that the level of miR-19b is significantly increased in patients with unstable angina, mechanistic studies were initiated in ApoE-deficient mice [137,138]. Levels of miR-19b were elevated systemically in the mice through intravenous injection of an endothelial cell-derived microparticle carrying the miRNA. The infusion of miR-19b promoted atherosclerotic changes in the ApoE-deficient mice. Carotid artery atherosclerosis progression was accelerated significantly through an increase in lipids, macrophages, and vascular smooth muscle cells (VSMC). It was also shown that miR-19b inhibited suppressor of cytokine signaling 3 expression (SOCS3) which is an important immunoregulator [138]. MiR-19b also induces dysfunction and apoptosis of cultured human aortic endothelial cells [139]. Bioinformatics predicted that JAZF1 would be a downstream target gene of miR-19b and a dual-luciferase gene reporter assay confirmed downregulation of JAZF1 in mouse arterial VSMC by miR-19b-3p. JAZF1 inhibition leads to atherosclerosis-promoting facilitation of vascular smooth muscle migration and proliferation [140].

Liao et al. induced myocardial infarction in C57BL/6 mice by coronary artery ligation and showed that a specific long coding RNA, BC002059, reduced infarct size and this reduction was prevented by miR-19b, which binds to and inhibits BC002059 [141]. Cell culture experiments showed that BC002059 suppresses the apoptotic process while miR-19b triggers apoptosis in cardiomyocytes. BC002059 and miR-19b act in opposition on the target gene α/β hydrolase domain containing (ABHD) 10 to prevent/promote apoptosis, respectively.

### 4.2. MiR-130a

Another miRNA of interest being explored through in vivo murine studies is miR-130a, which has been inversely associated with coronary heart disease in a multiethnic population in China [142]. In ApoE-deficient mice fed a high-fat diet, it was shown that miR-130a expression was elevated, as were serum levels of the inflammatory cytokines such as TNF-α, IL-1, IL-6, and IL-18. The relationship between miR-130a and cytokine levels was also documented in cultured human umbilical vein endothelial cells treated with LPS where miR-130a overexpression increased the levels of TNF-α, IL-1β, IL-6, and IL-18 while downregulation of this miRNA suppressed all four cytokines. Further establishing the atherogenic nature of miR-130a, its overexpression in human umbilical vein endothelial cells treated with LPS reduced the level of PPARγ and induced that of NF-κB [143].

### 4.3. MiR-10b

MiR-10b is known for its role in developmental regulation, angiogenesis, and cancer invasiveness [144,145]. However, miR-10b regulation of atherosclerosis is less well-explored. Bidzhekov and colleagues found elevated levels of miR-10b in human arteries containing atherosclerotic plaque compared to atherosclerosis-free arteries [146]. Hakimzadeh et al. found higher plasma levels of miR-10b in patients with chronic total occlusion undergoing percutaneous coronary intervention who had low versus high coronary collateral artery capacity [147]. This miRNA is also found at higher levels in blood plasma exosomes isolated from mice after wire injury of the femoral artery [148].

Studies have demonstrated that miR-10b functions in regulating proliferation of human VSMC [149,150]. It is also a crucial downstream regulator in the phenotypic conversion of human vascular endothelial cells from an anti-stenotic to a pro-stenotic phenotype, acting via suppression of latent-transforming growth factor (TGF) β-binding protein 1 (LTBP1). Nakahara et al. have shown in mice that LTBP1, a protein that fosters TGFβ-mediated control of smooth muscle proliferation, is produced by healthy anti-stenotic endothelium and transferred to smooth muscle via exosomes [148]. The knockout of miR-10b confers resistance to the development of atherosclerosis in high-fat, high-cholesterol diet-fed mice subjected to partial carotid ligation.

Wang et al. found that miR-10b directly modulates macrophage cholesterol transport via changes in expression of the ABCA1 and ABCG1 efflux proteins [151]. In cultured mouse peritoneal macrophages and in the THP-1 human macrophage cell line, transfection with miR-10b reduced cholesterol efflux and this effect was due to downregulation of ABCA1 and ABCG1. Transfection with miR-10b inhibitor sequence resulted in enhanced ABCA1 and ABCG1 expression. Another report by Wang et al. (2018) evaluated miR-10b expression in different stages of atherosclerosis and found variability in the arterial miR-10b levels that might partially result from the varying accumulation of apoptotic cells within atherosclerotic plaques. This causation was confirmed when the administration of antagmiR-10b brought about a reduction in advanced atherosclerotic plaque size and increased plaque stability in hypercholesterolemic ApoE-deficient mice. [152].

In contrast to other studies described here, Wu et al. found a beneficial function of miR-10b in the mouse heart under hypoxic conditions where its overexpression led to enhanced production of PTEN which, in turn, reduced cardiac apoptosis and preserved myocardium [153].

### 4.4. MiR-33

MiR-33 is an exosomal miRNA and one of the most abundant miRNAs in lipoprotein particles. [154,155]. MiR-33 is transcribed from an intron in the gene encoding the sterol regulatory element-binding protein (SREBP)-2 transcription factor [156,157]. SREBP-2 is a master regulator of cholesterol and lipid homeostasis and trafficking that acts by targeting the expression of the atheroprotective cholesterol transporter ATP-binding cassette subfamily A1 (ABCA1) [158]. Expression of miR-33 coordinates with SREBP-2 to promote lipid accumulation and one mechanism of this action is suppression of ABC transporters [159,160].

In murine macrophages, miR-33 deficiency improves reverse cholesterol transport, increases ABCA1 and ABCG1, and reduces lipid accumulation while also suppressing expression of pro-inflammatory genes involved in NFκB- and TLR4-mediated pathways [161]. Zhang et al. conjugated an anti-miR-33 sequence to a pH low-insertion peptide to achieve efficient delivery of the antisense oligonucleotide to atherosclerotic plaque macrophages of LDL receptor knockout mice fed a high-fat, high-cholesterol diet [162]. The treatment resulted in improved plaque regression, decreased lipid accumulation, and increased ABCA1 expression.

In a rat model of myocardial infarction, miR-33a was found to increase collagen deposition and myocardial fibrosis while inhibition of miR-33a via antagomiR-33a injection into the tail vein suppressed collagen formation and CF proliferation rate, whereas, in vivo inhibition improved left ventricular ejection fraction and attenuated cardiac fibrosis [163]. Complementary cell culture work showed that miR-33a was acting through the p38 MAPK signaling pathway to promote cardiac fibrosis.

Apart from directly antagonizing miR-33a, miR-33a inhibition effects can also be mimicked by selectively disrupting the interaction between miR-33a and its targets. Price et al. specifically demonstrated this by interfering with binding between miR-33a and ABCA1. They used CRISPR/Cas9 technology to generate mice in which the miR-33 binding sites within the 3′UTR of ABCA1 were edited to prevent miR-33 binding [164]. Macrophages from these mice have increased ABCA1 expression and cholesterol efflux capacity. Reconstitution of LDL receptor-deficient mice with bone marrow from mice with ABCA1 unable to bind miR-33 led to a reduction in atherosclerotic plaque formation and a decrease in foam cell formation. Similarly, in monkeys, targeting this miRNA increases plasma HDL and cholesterol efflux [165]. Although its ability to influence lipid metabolism and inflammation make it a promising candidate for therapeutic development, unfortunately, global inhibition of miR-33 in mouse models has harmful side effects such as cardiac dysfunction and promotion of obesity, insulin resistance, hepatic steatosis, and hypertriglyceridemia [161,166].

### 4.5. MiR-186-5p

MiR-186-5p was previously discussed in the context of its effect on VSMC in Section 3.2 [72]. Ding et al. found that serum exosomes isolated from humans within 72 h of onset of an acute myocardial infarction exhibited decreased levels of miR-186-5p compared to control exosomes [167]. One function of miR-186-5p is to suppress expression of the scavenger receptor lectin-like oxidized LDL receptor-1 (LOX-1) [168]. When cultured macrophages were exposed to serum exosomes from acute myocardial infarction patients, macrophage lipid uptake was enhanced due to elevated LOX-1 which occurred because the exosomes contained a dearth of miR-186-5p.

**Table 2 metabolites-13-00479-t002:** Exosomal miRNAs pertinent to cardiovascular function and disease.

MiRNA	Reference	Model Where Studied	Effect on the Cardiovascular System
MiR-19b	[137,138]	Elevated in persons with angina, further investigated in mice	↑atherosclerosis with ↑ lipids, ↑vascular smooth muscle, ↑ apoptosis
MiR-130a	[142,143]	Inverse association with coronary disease in humans, further investigated in mice and cultured human endothelial cells	Stimulates ↑ in serum levels of inflammatory cytokines
MiR-10b	[146,147,148,149,150]	Elevated level in plasma and in arteries in humans with atherosclerosis, further investigated in mice and cultured human endothelial cells and both mouse and human macrophages	Associated with pro-stenotic endothelial cell phenotype. Downregulates cholesterol efflux proteins ABCA1 and ABCG1 in macrophages.
MiR-33	[161,162,163,164,165,166]	Mice, rats and primates.	Potent inhibitor of ABCA1. Knockdown of miR33 in animal models increases HDL and enhances cholesterol efflux.
MiR-186-5p	[167]	Elevated in persons following acute myocardial infarction, further investigated in cell culture and mice	↓ levels in exosomes after myocardial infarction. Cultured macrophages exposed to these exosomes show ↑ scavenger receptor expression and ↑ cholesterol uptake

## 5. Leveraging Exosomes for CVD Treatment

Endogenous exosomes are sturdy and non-immunogenic and can transfer cargo from cell-to-cell across distances, thereby influencing the function and activity of the recipients. These properties can be used to advantage in CVD treatment to encapsulate specific therapeutic cargo such as miRNAs, mRNAs, and proteins for delivery to the heart and the concepts and strategies for accomplishing this will be expanded upon in this section as well as in Section 6 which is immediately follows [169,170]

The evolving understanding of the role of miRNA as an important contributing factor to cardiovascular development and CVD progression has brought this form of RNA to the forefront not only as a biomarker and diagnostic tool, but as a potential target for therapeutics for a wide range of cardiovascular disorders [11,171]. Therapeutic application entails manipulation of miRNA expression which, in turn, influences the miRNA–mRNA interaction and downstream pathways [172]. The ability of a single miRNA to control the translation of multiple mRNAs and of multiple miRNAs to regulate a single target gene is a caveat for unexpected consequences in changing miRNA expression as a treatment strategy.

The miRNA–mRNA interaction occurs post-transcriptionally at the 3′ untranslated region (UTR) of the target mRNA [173,174]. When an mRNA is targeted by a specific miRNA, it can lead to cleavage or inhibition of the translation of the mRNA sequence [175].

The generation of the mature miRNA strand begins with transcription of the miRNA from an intergenic, an intronic, or a polycistronic region of the genome by RNA polymerase II and, in some instances, RNA polymerase III, resulting in a long primary miRNA. This primary miRNA is then cleaved in the nucleus by the RNase III endonuclease Drosha, forming a precursor miRNA with a secondary hairpin-like structure. In humans, the cofactor, DiGeorge syndrome critical region 8 (DGCR8), is required for processing by Drosha [176,177]. The precursor is exported into the cytoplasm via the Exportin 5 (XPO5)/RAN-GTP complex. Secondary cleavage occurs once the miRNA is in the in the cytoplasm by RNase III endonuclease, also known as Dicer. This removes the hairpin loop to form a double stranded miRNA, which is subsequently uncoupled into a single guide strand by a member of the Argonaute (AGO) protein family and incorporated into an effector complex known as RNA induced silencing complex (RISC) [178,179]. RISC is the major mediator of miRNA–mRNA interactions. The miRNA-loaded RISC specifically recognizes target gene mRNAs by base-pairing and binding to the complementary region on the 3′ UTR of the mRNA. The complementary base pairing represses mRNA translation and/or induces mRNA cleavage, causing downregulation of expression of the protein encoded by the targeted mRNA.

MiRNAs are important in all stages of cardiogenesis and may act as biomarkers for heart defects [180]. Murine studies have brought about an understanding of the role of miRNAs in the development of the heart [181]. In juvenile mice, disruption of miRNA through the cardiac-specific targeted downregulation of Dicer leads to ventricular remodeling and enlargement, atrial enlargement, decreased cardiac function, and early death [182,183].

MiR-1 is highly enriched in the heart and represents about 24% of the miRNA in the human heart [184]. The miR-1 sequence derives from two almost identical transcripts (miR-1-1 and miR-1-2) and deletion of either of these in the mouse results in abnormalities in cardiomyocyte differentiation and conduction while knockout of both is lethal [185,186].

In contrast, the overexpression of certain miRNAs, in vitro, has led to the induction of concentric cardiac hypertrophy and heart failure [187,188]. This point shows the range of impact of miRNA on functionality of the heart. A number of miRNAs are necessary in maintaining heart health and disease prevention [189].

The central suppressive nature of most miRNAs involves impeding translation of proteins through the silencing of target mRNA sequences [190]. The short and simple sequence structure of miRNAs makes them viable candidates for therapeutic intervention to ameliorate cardiovascular disease. In general, exogenously induced overexpression of a specific miRNA will lead to greater suppression of translation of its target mRNAs, while underexpression of a miRNA will lead to increased translation of its target mRNAs [191]. The gain-and-loss-of-function mechanisms involved in the regulatory control exerted by miRNAs can potentially be applied using miRNA mimics to elevate beneficial miRNAs or miRNA silencers (antisense oligonucleotides, anti-miRs) to inhibit harmful miRNAs and each of these strategies could slow disease progression [192].

When naked RNA strands are introduced into the human body, they are rapidly degraded by ribonucleases in the blood. As a result, a number of methods have been tried in order to circumvent degradation [193]. MiRNA mimics are synthesized as double-stranded small RNA molecules that act in a way that approximates that of natural miRNAs [1757]. Anti-miRs are single-stranded oligonucleotides with sequence complementarity that allows for direct binding to the endogenous miRNAs of interest and block the miRNA-induced repression of mRNA translation through disruption of the miRISC complex [194]. Chemical modification of the oligonucleotide in the sugar ring and/or the backbone can increase stability and binding affinity without loss of functionality [195].

## 6. Exosome Delivery for Therapeutic Applications

In recent years, a shift from conventional delivery of drug therapy to more efficient drug delivery systems has been applied for a variety of diseases and disorders. This has occurred, in large part, due to the development of nanocarriers that can be used for delivery of not only drugs, but also of gene therapies [196,197,198]. Due to their unique characteristics, exosomes have become increasingly reliable and significant therapeutic biomaterials for use as nanosized carriers. Exosomes are able to distribute pharmaceutical compounds and bioactive materials to specific tissues and cells via their ability to transmit cargo to recipient cells through several mechanisms, such as surface receptor interaction, membrane fusion, and receptor-mediated endocytosis, phagocytosis, and/or micropinocytosis [199]. Their ability to act as targeted cargo delivery vehicles makes them useful in vaccine development, gene therapy, tissue regeneration, and cancer treatment [200,201].

The use of exosomes as bio-vectors in gene therapy has become increasingly appealing since they provide a high level of efficiency while remaining non-toxic [202,203]. Enhanced efficacy is a result of a combination of resistance to degradation and high-yielding, specific target cell uptake. Their lipid membrane protects exosome cargo from digestion, thus slowing clearance and their structural versatility allows for specific cellular targeting and controlled release of contents into the target cell [204].

In developing possible treatments for CVD via gene therapy, exosomes have shown great promise in animal models. For instance, Li et al. successfully applied exosomes in the treatment of familial hypercholesterolemia in an LDL receptor deletion mouse model [205]. They constructed LDL receptor mRNA-enriched exosomes that were stable and functional and injected them into LDL receptor-deficient mice fed a high-fat diet. The exosome encapsulated mRNA was translated into functional protein in the mouse liver, reducing hepatic lipid deposition and decreasing serum LDL cholesterol levels. In a study from Bouchareychas and colleagues, when exosomes isolated from cultured mouse bone marrow-derived macrophages (BMDMs) that had been polarized to an anti-inflammatory M2 phenotype were injected into atherosclerosis-prone Western diet-fed ApoE−/− mice, necrotic lesion areas of atheroma were significantly reduced [68]. The BMDM-derived exosomes were able to confer their anti-inflammatory properties on the mice, reduce levels of circulating neutrophils and monocytes and stabilize lesions despite continued Western diet feeding. In a further study, Bouchareychas and colleagues showed that the growth conditions of the BMDM determined the nature of their effect on atherosclerosis [206]. When the BMDM exosomes were grown under high glucose conditions to replicate a diabetes-type environment, they caused accelerated atherosclerosis when infused into ApoE−/− mice.

Schena et al. injected cortical bone stem cell-derived exosomes into a mouse model of ischemia-reperfusion injury immediately upon reperfusion and found that these exosomes reduced infarct size [207]. In companion cell culture studies of adult rat ventricular fibroblasts, the cortical bone stem cell-derived exosomes reduced markers of fibroblast activation.

Exosomes have also begun to be utilized in tissue regeneration. It is known that cardiac regeneration is possible because cardiomyocyte renewal has been demonstrated and adult human stem cells exist within endomyocardial biopsy specimens [208,209]. Further, intramyocardial injection of mesenchymal stem cells induced to acquire a cardiopoietic phenotype were beneficial in cardiac remodeling in patients with heart failure due to ischemic heart disease [210,211].

Exosomes secreted from mesenchymal stromal cells (MSCs) have shown a number of properties that make them capable of promoting cardiac repair [212,213,214]. MSC are a heterogeneous population of cells with tremendous plasticity and trilineage differentiation potential (osteogenic, adipogenic, and chondrogenic). MSC are readily accessible from bone marrow and adipose tissue and the exosomes they produce can then be isolated for study. In animal models, exosomes from MSC can be taken up by vascular endothelium and act to promote angiogenesis [215,216].

In a mouse model, MSC-derived exosomes harboring miRNA-185 were able to rescue cardiomyocytes from apoptosis after induction of myocardial infarction via coronary artery ligation [217]. In these mice, miRNA-185 was reduced in myocardial tissue after myocardial infarction and suppressor of cytokine signaling (SOCS)2, a target gene of miRNA-185, was increased. The addition of MSC-derived exosomes designed to overexpress miRNA-185 suppressed SOCS2, reduced apoptosis, and improved cardiac function. This anti-apoptotic effect on cardiomyocytes is also exhibited when human umbilical cord MSC-derived exosomes are injected into rats after induction of acute myocardial infarction [218]. By manipulating exosomal miR-19a content, the authors showed that protection of cardiac function by these exosomes was dependent upon their expression of miR-19a. They found that knockdown of SOX6, the target gene of miR-19a, was the mechanism of cardioprotection. SOX6 suppression leads to activation of the serine-threonine kinase AKT and inhibition of the c-Jun N-terminal kinases 3/caspase-3 pathway to apoptosis. Another mechanism through which infarct size and inflammation in the heart can be significantly alleviated by MSC-derived exosomes in mouse models is by shifting macrophage polarization in favor of the M2 phenotype over the M1 phenotype, thus regulating the immune microenvironment [219].

Despite these valuable effects, the efficacy of exosomes from natural and original MSCs remains limited because their expansion capabilities are minimal and their survival after transplantation is low, thereby leading to various efforts to optimize and engineer MSCs [220].

Exosomes isolated from cardiosphere-derived cells (CDCs), a type of cardiac progenitor cell (CPC) of intracardiac origin, have also been found to possess cardioprotective traits in animal models [221,222,223]. Among their many benefits include the ability to alleviate cardiac hypertrophy and fibrosis, reduce myocardial infarct size post-infarction, enhance endothelial cell tube formation, promote angiogenesis, and attenuate cardiomyocyte apoptosis [43,224,225,226,227,228]. Similar to MSCs, exosomes derived from engineered CDCs display an improved effect as well [229].

CPC-derived exosomes have also emerged as a promising candidate in therapy for cardiac repair by attenuating cardiomyocyte apoptosis, promoting tube formation by endothelial cells, and alleviating ischemic cardiac injury [230,231,232]. Exosomes from endothelial progenitor cells (EPC) may also have therapeutic value as shown in rodent models where they improve endothelial cell repair after balloon injury and protect against atherosclerotic injury of endothelium [233,234].

Exosomes are favorable options for therapeutic vehicles for functional cargo delivery considering their various intrinsic characteristics, including low immunogenicity and the ability to cross biological barriers. Additionally, their enhanced stability in various bodily fluids, including the digestive system, are particularly advantageous. They serve as delivery vehicles of incorporated drugs and other active compounds, which can act synergistically with the naturally occurring components of exosomes [196,235].

Although exosomes as nanocarriers hold great promise, a number of caveats in their utilization still need to be addressed. For instance, their clinical application can be limited due to their insufficient targeting ability [198]. Additionally, loading efficacy and capacity needs to be improved [236,237]. As not all the naturally occurring components of exosomes may be necessary for specific drug delivery, their constituents can be altered and modified in order to optimize them for their role as nanocarriers [238,239]. For instance, exosomal proteins can be modified with tannic acid so that the exosomes do not become adherent to blood vessel walls, allowing greater numbers to reach the myocardium where they can deliver their cargo [240]. Alternatively, exosomes can be conjugated with cardiac homing peptide which increased their uptake by cardiomyocytes in a rat model of ischemia/reperfusion injury [241].

Exosomes with or without modifications can be introduced into the body via multiple different administration routes, including intravenously, directly into the heart or coronary artery, orally, intradermally, and intraperitoneally. The route of administration of biomaterials directly affects their distribution within tissues [242]. Among the aforementioned routes, the most common is intravenous injection, but the disadvantage of this method is attenuated target delivery due to absorption by the liver, necessitating large doses of exosomes to compensate [33,202,243]. The intravenous delivery of exosomes is noninvasive and has been found in animal models to have beneficial effects in the repair of the heart after ischemic injury [35,244,245]. Absorption by the liver is avoided when exosomes are administered via an intracoronary or intramyocardial route, and exosomes delivered by these methods have been found to mitigate cardiac injury in animal models [246,247]. Another route of administration is via embedding exosomes in hydrogel cardiac patches that can then be placed on the heart or in the pericardial space during a minimally invasive surgical procedure [248].

Non-invasive external intervention using shock waves may stimulate crosstalk within the cardiac microenvironment [249]. In mice with myocardial infarction produced by coronary artery ligation, low energy shock waves promoted the release of angiogenic exosomes from endothelial cells in the ischemic myocardium. These exosomes contained miR-19a-3p, which stimulated angiogenesis and contributed to preserved left ventricular ejection fraction [250].

Despite the establishment of numerous commercial companies globally implementing the use of exosomes to deliver therapy, their utilization in CVD treatment is lacking. Most companies are focused on diseases such as cancer, neurodegenerative disorders, and genetic syndromes [196]. More intensive exploration is needed to help the millions of persons in need of cardiac functional improvement and prevention of myocardial ischemia and death.

## 7. Exosomes and miRNAs as Biomarkers of CVD

Endogenous exosomes are being studied as biomarkers for diagnosing and studying complex disorders because they represent a microcosm of their parental cell, contain components of that cell of origin, and reflect the pathological state of that cell [251,252,253] Exosomes of cardiac and vascular origin are easily accessible from blood, tissue, or body fluids and are considered to have great potential as biomarkers in the diagnosis of CVD and as predictors of future risk of myocardial infarction [254,255,256]. This field is in its infancy, awaiting further research to move into clinical applications [169,257,258,259].

Circulating levels of exosomal miRNAs have potential utility as prognostic indicators for heart failure and CVD. Wang et al. extracted exosomes from the plasma of 31 patients with acute heart failure and 31 controls and found that levels of two miRNAs that function as negative regulators of fibrosis, miR-425 and miR-744, were reduced in the heart failure patients [260]. Xiong et al. isolated exosomes from the plasma of persons with coronary heart disease confirmed by coronary angiography and found that, compared to controls, mRNA for sphingosine-1-phosphate receptor 5 (S1PR5) was upregulated and mRNA for carnosine synthase 1 (CARNS1) was downregulated and each were independent risk factors for coronary heart disease [21]. The authors suggest that these mRNAs might be useful as screening biomarkers for heart disease.

Problems arise in quantifying absolute values of circulating miRNA due to lack of standardized protocols, complicated multi-step procedures, and low sensitivity thresholds [253,261,262]. As new exosomal miRNA CVD biomarkers are validated, more uniform, rapid and economical procedures will be needed for clinical laboratory application.

## 8. Challenges in the Application of Exosomes in Diagnosis and Treatment

There are a number of obstacles in moving the use of exosomes from “bench-to-bedside” and among these are the lack of standardization of procedures involved in isolating and analyzing these microvesicles and the variation in yield that can result based on method of extraction, human error, storage conditions, and type of equipment [263]. The purity of exosomes and presence of contaminants may interfere with their characterization and quantification [264,265]. Another hurdle is the heterogeneous nature of exosomes which makes it difficult to reliably identify their cell of origin when extracting them from blood or body fluids, despite immunocapture technologies [266,267]. Another drawback is the high cost of analyzing exosome cargo [268,269].

Using exosomes in CVD therapy is fraught with its own issues, including unforeseen consequences of altering the microenvironment of the heart. Artificially constructed exosomal delivery vehicles may be immunogenic and may deviate from the route to the intended recipient cell type and cause undesirable off-target effects [270]. Exosomes cleared from the bloodstream can enter various organs, including the liver, spleen, and lungs [271,272]. Loading of specific RNA cargo into exosomes via the common methods of electroporation or sonication can be difficult and inefficient due to RNA degradation or aggregation [273]. The risk of infection is present when transferring these particles into the human circulation. Optimizing exosome dosage is difficult in light of the lack of uniformity in loading [274]. Establishing a correct therapeutic window is yet another requirement in human application [275].

## 9. Conclusions

Exosomes, biologically active nanosized lipid bilayer membrane vesicles, contain cargo derived from intracellular organelles of their cell of origin. Their contents, which include nucleic acids, proteins, and lipids, can be transferred locally as well as remotely to distant organs, enabling the exchange of information among cells. Their miRNA cargo is of particular interest because of the power of these molecules to regulate protein expression. Manipulation of miRNA has come into the spotlight as an expanding area of research through which CVD can be treated and prevented. Exosomal miRNAs transferred to cells of the heart and vasculature can influence important signaling processes that affect inflammation, lipid accumulation, angiogenesis, oxidative stress, and scar formation. The content of exosomes can also indicate the status of their cell of origin and may serve as a biomarker for monitoring cardiac damage or CVD risk. The ability of exosomes to deliver therapeutic agents directly to affected cells and tissues offers the promise of a targeted approach for treating CVD. The promising results from early animal studies and the potential for exosomes to revolutionize the treatment of CVD make it an exciting area of ongoing investigation. The move from preclinical evaluation to actual clinical use in patients will be challenging, but the continued morbidity and mortality from heart disease signals a need for breakthrough approaches to improve CVD outcomes. This is particularly true for high CVD risk patients with a prior history of myocardial infarction or angina and those with comorbidities such as diabetes, hypertension, obesity, and tobacco use. Medicine is moving toward precision treatment and designing exosomes to fit the needs of a particular patient may become possible with further intensive efforts.

## Figures and Tables

**Figure 1 metabolites-13-00479-f001:**
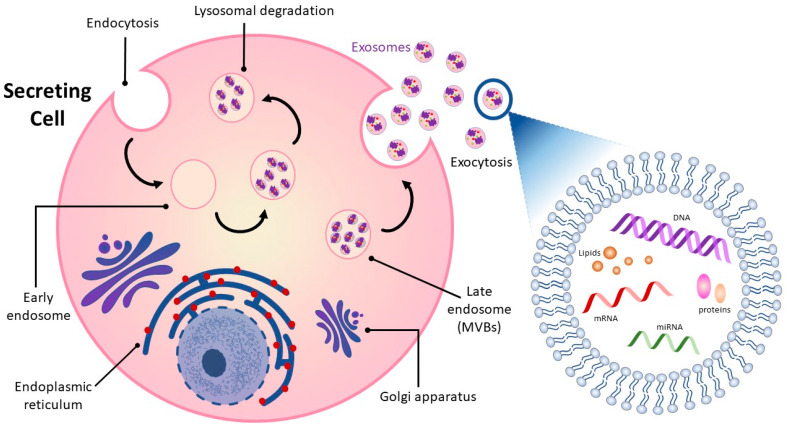
Exosome biogenesis and exosome cargo. Exosomes originate as intraluminal vesicles contained within multi-vesicular bodies (MVBs) and are formed via outward budding of the plasma membrane. Exosome content is cell-specific and contains noncoding RNA, such as miRNA and mRNA, lipids, and proteins. These contents play a vital role in intercellular communication as they reflect the pathophysiological state of the secreting cell.

**Figure 2 metabolites-13-00479-f002:**
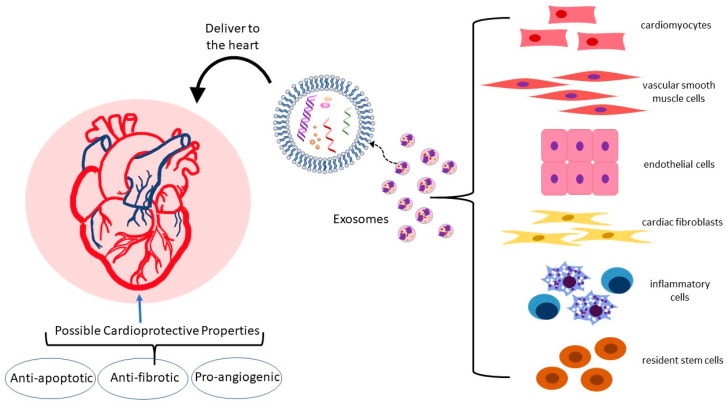
Exosomes and their cardioprotective potential. Exosomes derived from a variety of cell types can influence the heart via delivery of their RNA, DNA, lipid, and protein cargo. The exosomal contents can reach the heart via the bloodstream or directly through cell-to-cell contact and exert beneficial effects on apoptosis, fibrosis, and angiogenesis.

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
