# Peer review of "Exosomes in Cardiovascular Disease: From Mechanism to Therapeutic Target"

_metabolites, 2023, doi:10.3390/metabo13040479_

Round 1

Reviewer 1 Report

In this comprehensive review, authors have provided an excellent compilation of the association between exosomes and CVDs, with a specific focus on exosomal miRNAs. After a brief discussion on biogenesis of exosome, authors present the studies that reveal the association of exosomes in CVDs. Next, they focus on specific miRNAs identified in such exosomes and discuss their possible applications for therapeutic interventions. Later, the authors discuss the use of exosomes as a delivery vehicle followed by their potential as a biomarker.

Comments: Minor revision

1.      It would be useful for readers to have a separate section discussing challenges for application of exosomes as therapeutics and diagnosis.

2.      A table enlisting the CVD, cellular source of exosome and respective contents identified in those exosomes (specific protein/mRNA/miRNA) would provide more clarity in section 3.  

3.      Section 3.4 is mislabeled.

4.      Authors may cite following review of extracellular vesicles in CVDs: Cell Death Discovery volume 6, Article number: 68 (2020

Author Response

We thank the reviewer for thoroughly scrutinizing our manuscript. As requested, we have revised the manuscript and addressed the specific comments of each reviewer. The revised sections are delineated in red in a marked copy of the manuscript text.

Below, we provide a point-by-point response to each of the reviewer’s comments.

Reviewer 1:

Comment 1: It would be useful for readers to have a separate section discussing challenges for application of exosomes as therapeutics and diagnosis.

Response: We have added this section.

Comment 2: A table enlisting the CVD, cellular source of exosome and respective contents identified in those exosomes (specific protein/mRNA/miRNA) would provide more clarity in section 3. 

Response: We have added this Table (Table 1).

Comment 3: Section 3.4 is mislabeled.

Response: We apologize for the error and corrected the labeling to “Exosomes in Hypertrophy

Comment 4: Authors may cite following review of extracellular vesicles in CVDs: Cell Death Discovery volume 6, Article number: 68 (2020)

Response: We now include this reference (#169)

Reviewer 2 Report

Excellent report, comprehensive and approved for its publication. A reference table would be handy.

Author Response

We thank the reviewer for thoroughly scrutinizing our manuscript.

Below, we provide a point-by-point response to each of the reviewer’s comments.

Reviewer 2:

Comment 1: Excellent report, comprehensive and approved for its publication. A reference table would be handy.

Response: Thank you so much for this positive reception of our paper.

Reviewer 3 Report

The manuscript  entitled ‘Exosomes in Cardiovascular Disease: From Mechanism to Therapeutic Target’ resumes the current knowledge regarding the roles of exosomes in various cardiovascular ailments, as well as their therapeutic and diagnostic potential. The subject has expanded rapidly in the last decade and was recently addressed by a large number of comprehensive reviews (e.g. Extracellular vesicles: Targeting the heart - 10.3389/fcvm.2022.1041481; The Role of Exosomes and Exosomal MicroRNA in Cardiovascular Disease - 10.3389/fcell.2020.616161; Roles and Clinical Applications of Exosomes in Cardiovascular Disease 10.1155/2020/5424281; Characteristics and Roles of Exosomes in Cardiovascular Disease - 10.1089/dna.2016.3496; Extracellular vesicles in cardiovascular disease: Biological functions and therapeutic implications -10.1016/j.pharmthera.2021.108025; Small but significant: Insights and new perspectives of exosomes in cardiovascular disease - 10.1111/jcmm.15492; Extracellular vesicles in cardiovascular disease -10.1016/bs.acc.2020.08.006; Exosomes: Fundamental Biology and Roles in Cardiovascular Physiology - 10.1146/annurev-physiol-021115-10492).

Minor comments

1. Chapter 3.3 and 3.4 have identical titles, ‘Exosomes and apoptosis’ (correct 3.4. Exosomes and hypertrophy)

2. Chapter 4 is dedicated to miRNAs, a fundamental cargo of exosomes (‘Exosomal MiRNAs of Specific Interest in CVD’). Surprisingly, only four miRNAs are selected to be discussed here.

Line 347: ‘Here we have selected for discussion the miR- 347 NAs with the greatest potential clinical relevance in CVD’ - please clarify more extensively the selection criteria for miRNAs/ underline briefly the clinical superiority of the chosen examples

Line 349: ‘Human clinical trials are lacking, so much of the data is from murine studies’ – though, the table mentions results in humans (in 3 out of the 4 examples); please reformulate to avoid confusion.

Line 350: ‘We performed a literature review 350 using PubMed and Google Scholar, including the search terms “(atherosclerosis, atherosclerotic cardiovascular disease OR ASCVD) AND (obesity) AND (microRNA OR miR  OR miRNA)” from 2010 to the present, written in English-language journals’ – please justify in the text the association ATS – Obesity as search criteria for your miRNA selection, in a manuscript dedicated broadly to CVD.

3. Please insert references in Table 1.

4. Chapter 5: ‘Leveraging Exosomes for CVD Treatment’- please be so kind as to provide in a more direct manner a link between leveraging exosomes and the content of this section.

5. Chapter 7 – ‘Exosomes and miRNAs as Biomarkers of CVD biomarkers’ – there is a scarce approach of this generous topic; a table may be useful.

6. The limitations and controversies are occasionally discussed, but there is room for more.

7. Conclusions – they are too general, some leading ideas that may underline the originality of your manuscript among the great number of reviews published on the same subject would be salutary.

8. Acknowledgement – would be advisable to mention the reason why the authors are thankful to dr … and dr … .

Author Response

We thank the reviewer for thoroughly scrutinizing our manuscript. As requested, we have revised the manuscript and addressed the specific comments of each reviewer. The revised sections are delineated in red in a marked copy of the manuscript text.

Below, we provide a point-by-point response to each of the reviewer’s comments.

Reviewer 3:

Comment 1: Chapter 3.3 and 3.4 have identical titles, ‘Exosomes and apoptosis’ (correct 3.4. Exosomes and hypertrophy)

Response: We apologize for the error and corrected the labeling to “Exosomes in Hypertrophy

Comment 2a: Chapter 4 is dedicated to miRNAs, a fundamental cargo of exosomes (‘Exosomal MiRNAs of Specific Interest in CVD’). Surprisingly, only four miRNAs are selected to be discussed here.

Response: We have added a paragraph of description to explain our choices. We have added a fifth miRNA (miR-186-5p) to discussion and table with references.

Comment 2b: We Line 347: ‘Here we have selected for discussion the miR- 347 NAs with the greatest potential clinical relevance in CVD’ - please clarify more extensively the selection criteria for miRNAs/ underline briefly the clinical superiority of the chosen examples

Response: See answer to 2a - We have added a paragraph of description to explain our choices on page 10 and included new references as well.

Comment 2c: We Line 349: ‘Human clinical trials are lacking, so much of the data is from murine studies’ – though, the table mentions results in humans (in 3 out of the 4 examples); please reformulate to avoid confusion.

Response: We have deliberately chosen to highlight the exosomes where data is available in humans.

Comment 2d: We Line 350: ‘We performed a literature review 350 using PubMed and Google Scholar, including the search terms “(atherosclerosis, atherosclerotic cardiovascular disease OR ASCVD) AND (obesity) AND (microRNA OR miR  OR miRNA)” from 2010 to the present, written in English-language journals’ – please justify in the text the association ATS – Obesity as search criteria for your miRNA selection, in a manuscript dedicated broadly to CVD.

Response: We have now put in this explanation and been clearer about how we did the search. We did the search without obesity as well as with obesity and our wording was ambiguous so we apologize for this.

Comment 3: Please insert references in Table 1.

Response: We have added the references as suggested.

Comment 4: Chapter 5: ‘Leveraging Exosomes for CVD Treatment’- please be so kind as to provide in a more direct manner a link between leveraging exosomes and the content of this section.

Response: We have added a clarifying paragraph at the beginning of the section and also explained that the coverage of the topic continues in Section 6 that follows.

Comment 5: Chapter 7 – ‘Exosomes and miRNAs as Biomarkers of CVD biomarkers’ – there is a scarce approach of this generous topic; a table may be useful.

Response: Although we address the issue of biomarkers, this is not the primary focus of the manuscript. Our concentration is on mechanism and therapy and the paper is already very long with nearly 300 references. We have added reference 259 (Moreira-Costa, L.; Barros, A. S.; Lourenço, A. P.; et al. Exosome-Derived Mediators as Potential Biomarkers for Cardiovascular Diseases: A Network Approach. Proteomes 2021, 9, 8. doi: 10.3390/proteomes9010008) for those who wish to read in further detail.

Comment 6: The limitations and controversies are occasionally discussed, but there is room for more.

Response: We have added an entire new section “Challenges in the Application of Exosomes in Diagnosis and Treatment”

Comment 7: Conclusions – they are too general, some leading ideas that may underline the originality of your manuscript among the great number of reviews published on the same subject would be salutary.

Response: We have added more content to the conclusion, extrapolating to possible future use of exosomes in the clinical CVD setting.

  1. Acknowledgement – would be advisable to mention the reason why the authors are thankful to dr … and dr … .

Response: Done as requested.
